# The Association of Breakfast Frequency and Cardiovascular Disease (CVD) Risk Factors among Adolescents in Malaysia

**DOI:** 10.3390/nu11050973

**Published:** 2019-04-28

**Authors:** Norashikin Mustafa, Hazreen Abd Majid, Zoi Toumpakari, Harriet Amy Carroll, Muhammad Yazid Jalaludin, Nabilla Al Sadat, Laura Johnson

**Affiliations:** 1Centre for Exercise Nutrition and Health Sciences, School for Policy Studies, University of Bristol, Bristol BS8 1TZ, UK; norashikinmus@gmail.com (N.M.); z.toumpakari@bristol.ac.uk (Z.T.); hc12591@my.bristol.ac.uk (H.A.C.); 2Centre for Population Health (CePH) and Department of Social & Preventive Medicine, Faculty of Medicine, University of Malaya, 50603 Kuala Lumpur, Malaysia; hazreen@ummc.edu.my (H.A.M.); drnabilla@gmail.com (N.A.S.); 3Faculty of Public Health, Universitas Airlangga, Surabaya 60115, Indonesia; 4Department of Paediatrics, Faculty of Medicine, University of Malaya, 50603 Kuala Lumpur, Malaysia; yazidj@ummc.edu.my

**Keywords:** breakfast, obesity, cardiovascular, health, BMI, waist circumference, cholesterol, blood pressure, MyHeARTs

## Abstract

Breakfast frequency is associated with cardiovascular disease (CVD) risk in Western populations, possibly via the types of food eaten or the timing of food consumption, but associations in Malaysian adolescents are unknown. While the timing of breakfast is similar, the type of food consumed at breakfast in Malaysia differs from Western diets, which allows novel insight into the mechanisms underlying breakfast–CVD risk associations. We investigated foods eaten for breakfast and associations between breakfast frequency and CVD risk factors in the Malaysian Health and Adolescents Longitudinal Research Team study (MyHeARTs). Breakfast (frequency of any food/drink reported as breakfast in 7-day diet history interviews) and CVD risk factors (body mass index (BMI), waist circumference, fasting blood glucose, triacylglycerol, total cholesterol, high-density lipoprotein (HDL), low-density lipoprotein (LDL), and systolic and diastolic blood pressure) were cross-sectionally associated using linear regression adjusting for potential confounders (*n* = 795, age 13 years). Twelve percent of adolescents never ate breakfast and 50% ate breakfast daily, containing mean (SD) 400 (±127) kilocalories. Commonly consumed breakfast foods were cereal-based dishes (primarily rice), confectionery (primarily sugar), hot/powdered drinks (primarily Milo), and high-fat milk (primarily sweetened condensed milk). After adjustment, each extra day of breakfast consumption per week was associated with a lower BMI (−0.34 kg/m^2^, 95% confidence interval (CI) −0.02, −0.66), and serum total (−0.07 mmol/L 95% CI −0.02, −0.13) and LDL (−0.07 mmol/L 95% CI −0.02, −0.12) cholesterol concentrations. Eating daily breakfast in Malaysia was associated with slightly lower BMI and total and LDL cholesterol concentrations among adolescents. Longitudinal studies and randomized trials could further establish causality.

## 1. Introduction

Cardiovascular disease (CVD) is the number one cause of death globally [1]. In Malaysia, CVD is the biggest cause of mortality, accounting for 36% of total deaths [2]. Although the incidence of CVD (e.g., myocardial infarction and stroke) does not emerge until adulthood, CVD risk factors often present during adolescence [3]. For example, Malaysian children and adolescents have the highest prevalence of overweight and obesity across the South and Southeast Asian countries, at 22.5% (95% confidence interval (CI) 19.1−26.1) in boys and 19.1% (95% CI 16.1–22.6) in girls [4].

Over 50 cross-sectional studies, primarily in Europe and the United States, have reported consistent inverse associations between breakfast skipping and increased risk of overweight and obesity (odds ratio (OR) 1.55, 95% CI 1.46, 1.65) [5]. Outside of Europe and the United States, a meta-analysis of 19 cross-sectional studies in Asian populations also reported an increased risk of overweight and obesity (OR 1.76, 95% CI 1.48, 2.08) for breakfast skippers versus consumers [6], suggesting that the association is not limited to Western populations. Several causal mechanisms have been hypothesized to explain why eating breakfast may protect against CVD risk (Figure 1). In pathway 1, eating breakfast is hypothesized to reduce subsequent snacking, resulting in lower overall daily energy intake, thus maintaining energy balance and a lower body weight. Although in observational studies, breakfast is typically associated with higher total daily energy intakes [7]. In pathway 2, breakfast is associated with better food choices. For example, the kinds of foods typically eaten at breakfast in the United States and Europe tend to have cardio-protective properties e.g., wholegrain cereals that are high in fiber and micronutrients [8]. However, as foods typically eaten for breakfast vary widely across cultures, consistent associations of breakfast frequency across diverse countries could point to the timing of eating as more important than the type of food eaten. In pathway 3, it is hypothesized that eating in the morning is specifically better suited to circadian rhythms in metabolism, such that food ingested earlier in the day is metabolized more efficiently. The hypothesized circadian pathway is supported by acute randomized crossover experiments in adults, whereby the timing of food intake (but not the type of food consumed) is manipulated such that delaying breakfast is related to poorer appetite control, lower resting energy expenditure, impaired fasting lipid profiles, and reduced postprandial insulin sensitivity [9,10,11,12]. However more recent longer-term free-living trials that manipulated the timing of the first meal or randomized participants to receive simple advice to “eat breakfast” or not, have cast doubt on whether observational associations are causal but instead confounded i.e., eating breakfast may simply be an indicator of a generally healthy lifestyle [13,14,15]. Further research exploring breakfast frequency and the types of food eaten for breakfast in non-western populations is required to further understand existing observed associations.

Most research on breakfast and health has been conducted in adults, with some work in children, but far less research has been dedicated to adolescents. However, adolescence represents a unique phase of life involving rapid growth, hormonal fluctuations, insulin resistance [16,17], and circadian dysregulation [18]. Considering the insulin resistance and circadian dysregulation can be influenced by breakfast consumption [19], it is important to understand the role of breakfast consumption on health in adolescents. In children aged 9–10 years in the 2011–2013 multi-country ISCOLE study, eating breakfast frequently (6–7 days/week), which varied from 58% in Brazil to 95% in Colombia, was associated with lower body mass index (BMI) z scores in some but not all countries [20]. This observation is supported by a systematic review including 10 longitudinal studies of children and adolescents of which eight provided evidence of inverse associations of more frequent breakfast consumption and adiposity [21]. In Malaysia, four cross-sectional studies have identified breakfast as the most frequently skipped meal among children and adolescents [22,23,24,25], with the prevalence of skipping breakfast varying from 17% [25] to 40% [24]. Only one cross-sectional study has examined breakfast consumption in relation to adiposity and obesity among Malaysian 12–19 year old adolescents [26]. Adolescents consuming breakfast five or more days/week had lower body weight, BMI, waist circumference, body fat mass, and percentage of body fat compared to those consuming breakfast less than five days/week. Overall, observational evidence on the association of a higher breakfast frequency with lower adiposity in adolescents is consistent worldwide. 

Given that higher adiposity is associated with poorer CVD risk profiles [27], and breakfast frequency is associated with adiposity, one might expect to see similar associations with breakfast frequency and cardiovascular risk. However, evidence for associations of breakfast consumption with other CVD risk factors is less consistent. For example, fasted blood glucose concentrations were lowest among U.K. adolescents [28] and European boys [29] reporting daily breakfast consumption, but there was no evidence of a similar cross-sectional association among U.S. [30], Finnish [31], Taiwanese [32], or Japanese [33] adolescents. Lower systolic blood pressure has been observed among daily breakfast consumers in Taiwan [32] and in a European sample of boys, but no evidence of a similar cross-sectional association was observed among European girls and U.K., Japanese [33], U.S., Finnish, or Iranian [34] samples. Among Australian 9–15 year old children and adolescents followed-up as adults (aged 26–36 years), persistent breakfast skipping (defined as not eating between 0600 h and 0900 h as a child and adult) compared to consuming breakfast consistently was longitudinally associated with a higher waist circumference (mean difference 4.6, 95% CI 1.72, 7.53 cm), fasting serum insulin (mean difference 2.02, 95% CI 0.75, 3.29 mU/L), total cholesterol (mean difference 0.40, 95% CI 0.13, 0.68 mmol/L), and low density lipoprotein (LDL) cholesterol (mean difference 0.40, 95% CI 0.16, 0.64 mmol/L) concentrations [35]. Among Swedish 16 year-olds followed-up to age 43 years, eating breakfast only showed robust evidence of association with hyperglycemia (OR 1.75, 95% CI 1.01, 3.02), but not low high density lipoprotein (HDL) concentrations or hypertension [36]. Differences in sample sizes (ranging from 367 to 13,486 participants), diverse definitions of breakfast habits, and variation in adjustment for confounders may explain the inconsistency in findings to date. 

Another possible explanation for inconsistent associations between breakfast consumption and CVD risk in adolescents is a lack of concurrent information on breakfast composition or diet quality. Existing evidence has typically relied on simple self-reported questions on usual breakfast frequency leaving it unclear how regular breakfast consumption relates to the types of food specifically eaten in the morning. Detailed dietary assessment methods like a food diary, 24-h diet recall, or diet history provide data on both the time of consumption and the food and nutrient content of breakfast, enabling both the frequency and composition of breakfast to be characterized and separate possible mechanisms to be explored. Finally, a proposed method for improved causal inference in observational research, by reducing the possibility that associations are confounded, is to explore the consistency of associations across high, middle- or low-income countries where the confounding structure may differ [37]. Therefore, adding to the existing evidence base on associations between breakfast and CVD risk, that is primarily in Western populations, by using Malaysian data may improve our understanding of the importance of breakfast.

No study to date has investigated the association of breakfast and CVD risk in Malaysian adolescents. In addition, few previous studies were able to explore what is eaten for breakfast in addition to breakfast frequency. The purpose of our study was to provide a detailed description of what adolescents ate for breakfast and investigate the association of breakfast frequency with CVD risk factors among adolescents using data from the Malaysian Health and Adolescent Longitudinal Research study (MyHeARTs), including 7-day diet history enabling adjustment for potential confounders including concurrent diet quality.

## 2. Methods

### 2.1. Study Overview and Population

MyHeARTs is a prospective cohort study of 1351 schoolchildren (aged 13 years old at baseline) that attended 15 public secondary schools from central (Kuala Lumpur and Selangor) and northern (Perak) regions of Peninsular Malaysia. A detailed description of the MyHeARTs can be found elsewhere [38]. All procedures in MyHeARTs were approved by the Medical Ethics Committee, University Malaya Medical Centre according to the Declaration of Helsinki. Participation in the study was voluntary and written informed assent and consent was obtained from participants and parents/guardians of participants, respectively. 

### 2.2. Dietary Assessment 

Dietary intake was assessed using a seven-day diet history, which has been shown to give the most valid estimates of energy intake in children and adolescents (Burrows et al. 2010). Dietitians conducted open-ended interviews with the students to collect information on foods and drinks consumed during breakfast, mid-morning snack, lunch, afternoon tea, dinner and supper over the previous seven days. Pictures of local food and household measurement items were used to assist participants in estimating portion sizes consumed. Food and drink energy and nutrient intakes were calculated using the Nutritionist Pro™ Diet Analysis (Axxya Systems, Redmond, WA, USA) and Nutrient Composition of Malaysian Foods [39,40].

To compute breakfast consumption, energy intake from all foods or drink reported during the meal-slot identified as breakfast was combined on each day. Frequency of breakfast intake (days/week) was calculated as the number of days where any food or drinks were reported at breakfast. This was categorized for descriptive analyses by grouping into daily, 4–6 days/week, 1–3 days/week, and 0 days/week. The types of food and drink reported were investigated by grouping food items into 44 groups and then analyzing the frequency that each food group was reported across all food items in each breakfast occasions. Under-reporting of energy intake was assessed by comparing reported energy intake to estimated basal metabolic rate and participants defined as under-reporters (*n* = 497) were excluded [41]. Excluded participants were 43% male, 57% urban, 78% Malay, 12% Chinese, 7% Indian, and 2% other ethnicity, with age mean 12.9 (±0.3) years, BMI mean (±SD) 22.2 (±5.6) kg/m^2^, and a mean daily energy intake of 1126 (±278) kcal.

A dietary pattern score for each adolescent was derived using reduced rank regression [42]. The score is a weighted linear combination of the 44 food groups (Appendix A) which predict the maximum variation in intermediate variables hypothesized to be on the pathway between food group intake and cardiovascular health, i.e., energy density (kJ/gram), fiber density (g/MJ), and percentage of energy intake from fat (%). A high pattern score represents an energy dense, low-fiber diet, which is characterized by high consumption of processed meat, bread products, chocolate, egg dishes and energy-dense Malaysian kuih (Malaysian snacks and desserts), and a low consumption of fruit, vegetables and vegetables dishes, and soups (unpublished results). 

### 2.3. Anthropometric and Clinical Measurements

Height was measured without socks and shoes using a calibrated vertical stadiometer (Seca Portable 217, Seca, Birmingham, UK). Weight was measured with light clothing using a digital electronic weighing scale (Seca 813, Seca, Birmingham, UK). Body mass index was calculated as weight in kilograms divided by the square of height in meters (kg/m^2^). Waist circumference was measured to the nearest millimeter using a non-elastic Seca measuring tape (Seca 201, Seca, Birmingham, UK). Systolic and diastolic blood pressure were calculated as the average of three readings using a stethoscope and a mercury sphygmomanometer. Fasting (≥10 h) venous blood samples were obtained from each participant and analyzed for serum total cholesterol, LDL cholesterol, HDL cholesterol, triacylglycerol (TAG), and blood glucose concentrations. More information on the way anthropometric and clinical measures were obtained can be found elsewhere [38].

### 2.4. Confounders 

Physical activity was self-reported using the physical activity questionnaire (in Malay) for older children (PAC-Q). Sex, ethnicity (Malay, Chinese, Indian, other), smoking status (yes/no), and alcohol intake (yes/no) were self-reported in student questionnaires. 

### 2.5. Statistical Analysis 

Variables are described using mean (SD) if continuous and normal, or frequency (%) if categorical. Normality was checked via visual inspection of histograms. Analysis of variance (ANOVA) was performed to test the mean difference between breakfast frequency groups and continuous CVD risk factors. Tests for linear trend were performed using a likelihood ratio test comparing a model containing breakfast frequency (days/week) as a continuous variable with a model containing breakfast frequency as a categorical variable (daily, 4–6 days/week, 1–3 days/week, and 0 days/week). The chi-squared test was used to test for associations between breakfast frequency groups and categorical confounders. Multiple linear regression analysis was used to examine the independent relationship between breakfast frequency (days/week) and CVD risk factors. Risk factors analyzed as dependent variables were BMI, waist circumference, fasting serum glucose, TAG, total cholesterol, HDL and LDL concentrations, and systolic and diastolic blood pressure. Model 1 included only breakfast frequency as an independent variable. Model 2 included breakfast frequency, sex, ethnicity, physical activity level, smoking status, and alcohol intake. Model 3 included model 2 variables as well as BMI. Model 4 included model 3 variables together with total energy (kcal/day), total carbohydrate (% daily energy), total protein (% energy), total fat (% energy), total cholesterol (mg/1000 kcal), total saturated fatty acid (% energy), total sodium (mg/1000 kcal), total calcium (mg/1000 kcal), total iron (mg/1000 kcal), total fiber (g/1000 kcal), and total sugar (% energy). Model 5 included model 4 variables together with dietary pattern score. Statistical analyses were performed using SPSS (version 21, IBM), Stata (version 15, StataCorp LLC, College Station, Texas, USA), and SAS (version 9.4, Marlow, Buckinghamshire, UK). 

## 3. Results

Figure 2 shows that 795 participants had complete data based on the variables of interest and were included in the current analysis. The characteristics of participants overall and stratified by breakfast frequency are presented in Table 1. Fifty one percent of the sample were daily breakfast eaters, 25% consumed breakfast 4–6 days/week, 15% consumed breakfast 1–3 days/week, and 10% of the sample never consumed breakfast. Adolescents of Indian ethnicity were more likely to be daily breakfast eaters (71%) compared to the overall sample, while no Chinese adolescents were identified as breakfast skippers. Boys were more likely than girls to consume breakfast daily (58% vs. 47%). Few differences in breakfast frequency were observed for any other individual characteristics. 

In terms of diet, adolescents who consumed breakfast but infrequently (1–3 days/week) consumed the least total daily energy, calcium, and sugar compared to both breakfast skippers and daily consumers (Table 1). Daily breakfast consumers had a positive dietary pattern score i.e., a more energy dense and lower fiber density diet, compared to infrequent breakfast eaters and breakfast skippers who, on average, had a negative dietary pattern score. Breakfasts reported in MyHeARTs contained a mean (SD) 400 ± 127 kcal. The most frequent food groups of all food items reported at breakfast were cereal-based mixed meals (19%), chocolate and confectionery (14%), hot and powdered drinks (13%), high fat milk (12%), and bread (6%) (Figure 3). Cereal-based mixed meals, were mainly rice in coconut milk (34% of cereal based mixed meals food items), fried rice (30%), and fried noodles (12%) (Appendix A). Chocolate and confectionery refer to granulated sugar (99% of chocolate and confectionery items reported at breakfast), while hot and powdered drinks, are mostly Milo (a chocolate and malt powder drink) (40%) and malted milk powder (38%). High fat milk and cream was primarily sweetened condensed milk (90% of items in that group) (Appendix A).

### 3.1. Mean of Cardiovascular Disease Risk Factors 

A description of CVD risk factors by breakfast frequency is presented in Table 2. Compared to daily breakfast consumers, those who never ate breakfast had slightly higher serum total cholesterol concentrations (4.6 vs. 4.8 mmol/L, *p* = 0.01), driven by higher LDL cholesterol (2.7 vs. 2.9 mmol/L, *p* = 0.01) (Figure 4). These associations were linear across the breakfast categories. Unlike the other CVD risk variables, there was weak evidence (*p*_trend_ = 0.06) that the relationship between breakfast consumption and BMI may be non-linear. Those consuming breakfast daily had the lowest BMI (19.2 kg/m^2^), but participants who consumed breakfast infrequently (1–3 days/week), rather than never, had the highest BMI (20.9 kg/m^2^; Table 2 and Figure 4). There was no evidence of other CVD risk factors, i.e., fasting serum glucose, TAG, or HDL concentration and blood pressure, varying by breakfast frequency, though there was some weak evidence suggesting that waist circumference was 2 cm smaller in daily vs. never breakfast consumers (*p* = 0.06). 

### 3.2. Association of Breakfast Consumption and cardiovascular disease (CVD) Risk Factors 

Table 3 shows associations between breakfast frequency and CVD risk factors, before and after adjusting for potential confounders. In the fully adjusted models, each extra day of breakfast was associated with a lower BMI by 0.2 kg/m^2^ (95% CI −0.4, −0.1) and lower total cholesterol concentration of −0.03 mmol/L (95% CI −0.06, −0.01). The lower total cholesterol concentration was driven by LDL cholesterol, which was lower by the same order of magnitude for each extra day of breakfast (β −0.03, 95% CI −0.06, −0.01). Waist circumference was 0.5 cm (95% CI −0.84, −0.16 cm) lower per extra day of eating breakfast per week in model 2, but the association was reduced to 0.06 cm/breakfast/week after adjustment for BMI (model 3, Table 3). No evidence of association was observed for fasting serum glucose or TAG concentration, serum HDL cholesterol concentration, and blood pressure.

## 4. Discussion

In this cross-sectional analysis of a large sample of Malaysian adolescents with detailed dietary data and a broad range of CVD risk factors, we found evidence that breakfast frequency was inversely associated with BMI and fasting total and LDL cholesterol concentrations, but not waist circumference, fasting glucose, HDL or TAG concentrations, or blood pressure. 

Among Malaysian adolescents in our study, 10% were breakfast skippers and 51% were daily breakfast consumers, which is similar to a previously reported prevalence in a different Malaysian sample of 12–19 year-olds [26]. A higher prevalence of breakfast skipping has been observed among U.S. and Finnish adolescents (i.e., 13–16% and 16–24% respectively [31,43]) as well as higher prevalence of daily breakfast consumption in other populations (i.e., 94% among U.K. primary school children and 89% among Taiwanese primary school children [28,32]). Differences in daily breakfast consumption could partly be explained by a lack of a consistent definition of breakfast in the literature [44]. For example, some studies defined breakfast based on the time of the day food was consumed, e.g., between 0600 h and 0900 h [35], while others based their definitions on the frequency of breakfast (self-identified by the participant) during the week [29,32,34] or only during weekdays [31]. In addition, adolescents with the same breakfast frequency may be in different groups across studies, e.g., breakfast skippers in the current study are those who never consumed breakfast, while in other studies those who never consumed breakfast are combined with irregular consumers [29]. 

### 4.1. Breakfast Associations with Cardiovascular Disease Risk Factors

Our findings agree with previous observational studies across the world showing consistent inverse associations between frequency of breakfast consumption and BMI [5,6]. Only one study previously investigated Malaysian adolescents (*n* = 236) [26], finding breakfast consumption to be associated with lower BMI, concordant with our analysis. Interestingly, we did not find strong evidence for a linear relationship potentially suggesting that never eating breakfast is not problematic per se, but rather irregular consumption of breakfast on some days but not others makes energy balance more difficult to regulate. Nurul-Fadhilah et al. [26] further found an inverse relationship between breakfast consumption and waist circumference, which our analysis supports, but the association was not robust to adjustment for BMI. Similar waist circumferences were found between the two studies for infrequent breakfast consumers (~70 cm) but the waist circumference observed in frequent breakfast consumers differed, where Nurul-Fadhilah et al. [26] found lower waist circumferences (~64 cm) compared with MyHeARTs (~68 cm). The differences in these findings may be explained by the comparatively homogenous sample (mostly Malay from Kelantan) with a smaller sample size compared to the larger and more diverse MyHeARTs sample.

Our study found frequent breakfast consumption was associated with lower total cholesterol driven by lower LDL cholesterol concentrations compared to infrequent breakfast consumption. These findings are in agreement with some [29,35], but not all [28,29,30,33] previous studies. However the effect sizes in our analyses were small and likely not clinically meaningful (~0.03 mmol/L difference in plasma total and LDL cholesterol concentrations for each extra day of breakfast), as ≥1.0 mmol/L reduction in LDL concentration is needed to reduce the risk of all-cause mortality by 10% in adults [45]. 

Previous research has reported inconsistent associations between breakfast consumption and HDL cholesterol concentrations [30,31,32,33]. Our analyses however provided no evidence of association in accordance with several studies [30,31,33,35]. Whilst others found statistically significant associations [32,46], the differences in HDL cholesterol concentrations observed elsewhere were small and unlikely to be clinically meaningful at reducing CVD risk (difference of <2 mg/dL). Importantly, all categories of breakfast consumption in this study were associated with plasma HDL cholesterol concentrations within the range associated with lowest CVD and mortality risk in adults [47]. Such findings may suggest that there is limited power in detecting associations between CVD risk markers such as HDL concentrations in an overall healthy sample. Despite the inconsistencies in statistical significance between studies, taken together there appears to be no meaningful association between HDL cholesterol concentrations and breakfast consumption in adolescents, in line with our results. 

We did not observe evidence of association between breakfast consumption and blood pressure, which may partially be explained by the similar sodium intakes between breakfast consumption categories. As with HDL cholesterol concentrations, previous research has provided mixed results regarding the relationship between breakfast consumption and blood pressure in adolescents. Ahadi et al. [34] found breakfast consumption to be associated with lower blood pressure, supporting previous research showing a reduced risk of hypertension [31]. The latter study however only showed lower hypertension risk in girls who were semi-regular breakfast consumers. Our findings do however support other research demonstrating no association between breakfast consumption and diastolic [28,30,36,46] and systolic [30,36,46] blood pressure. 

Our analysis did not find an association between breakfast consumption frequency and plasma TAG concentrations, supporting most other studies [28,29,30,48]. Smith et al. [35] found breakfast skipping to be longitudinally associated with increased serum TAG concentrations compared to regular breakfast consumption, potentially suggesting changes to TAG concentrations occur over a longer period. Such a time effect may be physiologically plausible; blood TAG concentrations are typically due to greater adiposity, but such an effect is not necessarily immediate. Since breakfast skipping has consistently been associated with greater adiposity [5], it may be that the associations with TAG are chronic adaptations to higher fat mass. 

Lower fasting glucose concentrations have previously been associated with regular breakfast consumption in adolescents in some [28,29,36] but not all [30,32] studies, with others finding mixed results [31,35]. Our analyses provide no evidence for an association. However, we were unable to investigate markers of insulin resistance (e.g., Homeostatic model assessment HOMA) due to lack of insulin data; such a measure provides useful data due to the insulin resistance exhibited during adolescence [16,17]. This may explain the more consistent finding (compared to fasting plasma glucose concentrations alone) of regular breakfast consumption being associated with lower insulin resistance in adolescents [28,29,30,35]. 

There is a lack of causal data in adolescents pertaining to glycemic regulation and breakfast consumption. In adults, short-term experimental manipulations to meal regularity (including breakfast omission) have detrimental effects on acute postprandial glycemic and insulinemic responses [9,49], although longer-term breakfast consumption versus omission does not appear to negatively influence glycemic regulation [13]. Equally, in schoolchildren, fasted plasma glucose concentrations were unaffected by participation (compared to non-participation) in a nine-month school breakfast program [50], supporting our analysis. Such findings suggest chronic adaptations to blood glucose possibly occur mitigating the acute-effects of breakfast omission. 

### 4.2. Comparisons of Breakfast Definitions

The differences in the way breakfast is defined and how frequencies are combined in each study make it hard to directly compare associations with health and may be responsible for the disparate findings [51]. For example Nurul-Fadhilah et al. [26] used two broad categories of breakfast, i.e., <5 days/week versus ≥5 days/week, which could explain the evidence for association observed for waist circumference, in comparison to the weaker evidence observed in our study, which included a greater number of breakfast categories. Furthermore, our study adjusted for aspects of adiposity and diet quality, which was not the case in previous studies showing associations for fasting glucose concentrations and blood pressure [28,32]. Different ranges of CVD risk factors could also explain the diverse findings; for example, mean diastolic blood pressure was higher among U.K. primary children [28] compared to our sample. Additional explanations could be the inclusion of younger and smaller sample sizes [28,30] and the inclusion of healthier adolescents compared to the average population [33]. 

### 4.3. Breakfast Composition and Meal Timing

Overall, our findings show some, albeit limited, evidence for a protective association between breakfast consumption and some CVD risk factors, such as BMI, and total and LDL cholesterol concentrations. One reason for which breakfast might be protective is the consumption of particular foods during breakfast. The fiber contained in cereal and the high protein content of eggs have been shown to increase satiety and improve appetite control [44,52,53]. This may explain the protective effect of breakfast in some populations. For example, European children and adolescents typically consume micronutrient-dense breakfast cereal (with milk) and bread [54], while in our sample the most commonly consumed breakfast foods were cooked rice dishes, sugar, malt drinks, and sweetened condensed milk. Considering the differences in the types of foods consumed at breakfast between populations, it is more likely that, if causal, breakfast relates to health because of the timing of food intake in relation to circadian metabolic rhythms rather than the type of food consumed. 

The timing of eating has been associated with improved gluco-regulatory responses in adults (e.g., Farshchi et al. [9]). Improved postprandial metabolic responses may be of particular importance in adolescents because of the increased insulin resistance in this population compared to childhood and adulthood [16,17], as well as the circadian dysregulation, which often skews adolescents towards evening chronotypes [18] typically associated with poorer general health [55]. In adults, such circadian misalignment is causally implicated in poorer cardiometabolic health (e.g., Scheer et al. [56]), with observational research suggesting similar health outcomes in adolescents [57]. Compared to the evening, morning ingestion of nutrients results in lower postprandial glycemia and insulinemia, conducive to lower CVD risk [58], making breakfast a particularly important meal. Breakfast consumption may therefore act to improve cardiometabolic health via both circadian entrainment [59], and via exploiting the favorable postprandial response found when ingesting nutrients in the morning compared to the evening [60], though causal research specifically in adolescents is lacking and should be a priority for future studies. 

### 4.4. Strengths and Limitations

One of the main strengths of our study is that the MyHeARTs is the largest study in Malaysia, involving adolescents from three main ethnicities, three regions, and the inclusion of rural and urban participants, providing superior representativeness than previous studies (e.g., Nurul-Fadhilah et al. [26]). The ethnic diversity in our analyses is a particular strength, demonstrating higher breakfast eating prevalence among Indian participants, and no cases of skipping breakfast every day in Chinese participants. Dietary intake was measured using a seven-day diet history, which is a valid method for estimating the energy and nutrient intake among adolescents (Burrows et al. 2010). A broad range of demographic data, with student lifestyle and information on parents enabled us to model the independent association of breakfast frequency with CVD risk factors, including multiple biomarkers. 

Whilst the strengths of this study mean we have provided novel data on breakfast and cardiometabolic health in Malaysian adolescents, several limitations also need to be acknowledged. Breakfast is notoriously difficult to define, particularly in epidemiological studies due to limitations in data availability (e.g., wake time not being recorded). In our study, breakfast was self-defined by the participant. Recent research has aimed to define breakfast and has proposed a wake-time-dependent definition of within two or three hours of waking [44]. In data collection for MyHeARTs, dietitians asked participants when their breakfast time was, and if this was after 1030 h, the meal was not recorded as ‘breakfast’ but as a morning snack instead. Owing to early school start times in Malaysia (~0730 h), our definition of breakfast on week days is likely to have classified some breakfasts outside of 2–3 h of waking, thus capturing a slightly extended morning fast. 

Future studies should further explore different types of breakfast which may help provide novel insights into the role of breakfast on cardiometabolic health. For example, in our sample, adolescents typically consumed foods such as rice and noodles for breakfast which, despite being high carbohydrates, differ in nutritional composition to typical breakfast foods in Europe, such as cereal, which is a vehicle for milk consumption in European children [54]. Differences in breakfast composition may help explain some of the aforementioned disparities compared to previous research. Future studies could explore this using meal coding, as has been done previously, to understand meal-based dietary patterns in adults [61,62]. Additionally, understanding breakfast composition may help elucidate whether the consistent associations with BMI are driven by breakfast food composition or timing of nutrient intake. In the case of the latter, this would explain why we observed a protective association with breakfast despite daily breakfast consumers having a more energy-dense, lower fiber overall dietary pattern score, which is associated with greater adiposity in the United Kingdom [63]. This may additionally suggest further cultural differences, as typically breakfast consumption in Western populations is associated with less obesogenic diets [64]. 

Physical activity and diet were both self-reported. Although validated methods were used to obtain these data minimizing risk of bias, there is known error within these measures, such as energy intake underreporting bias. As such, we aimed to control for dietary misreporting by excluding under-reporters. Finally, the cross-sectional and observational nature of the study prevents causal inference for the associations observed owing to the possibility of reverse causation and residual confounding. Thus, both longitudinal and randomized controlled studies are required in order to establish the causality of the relationship between breakfast consumption and lower CVD risk factors in this population. Longitudinal studies are particularly valuable in adolescents as some research has suggested breakfast skipping is cross-sectionally, but not longitudinally associated with higher BMI in adolescents with overweight, and vice versa in those in the normal BMI range [65]. Our study has provided valuable information for hypothesis generation to aid in developing such studies. 

## 5. Conclusion

In conclusion, the present study suggests that adolescents in Malaysia who eat breakfast more frequently have a lower BMI, and lower plasma total and LDL cholesterol concentrations independent of a range of confounders. Our study results specifically highlight that the benefits of breakfast (if causal) are not limited to the types of foods typically eaten in Western populations such as breakfast cereals, milk, or bread, but may be independent of the type of food consumed and instead related to the timing of food intake. Furthermore, our data suggested that irregular habits may be more detrimental to CVD risk than uniform breakfast eating vs. skipping. Previous longitudinal studies or long-term randomized trials have not explored the potential impact of consistent versus irregular breakfast habits or been able to differentiate changes in the timing of eating from the type of food eaten. Future research should aim to examine whether the content, timing, or regularity of breakfast intake is associated with lower CVD risk both longitudinally and causally within trials in adolescents.

## Figures and Tables

**Figure 1 nutrients-11-00973-f001:**
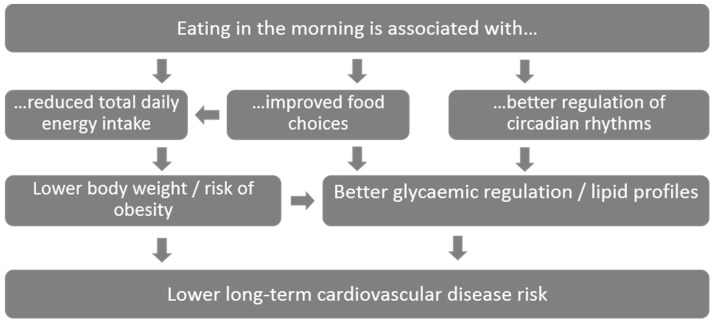
Hypothesized mechanisms for associations of breakfast with cardiovascular disease risk.

**Figure 2 nutrients-11-00973-f002:**
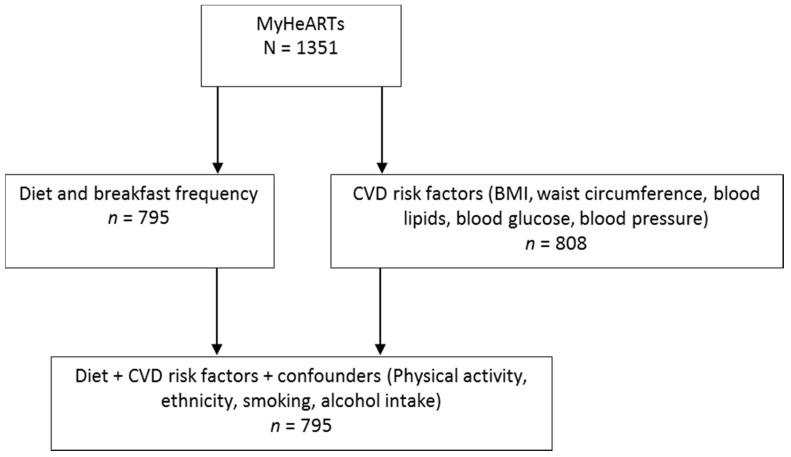
Number of participants with data available based on the variables of interest. CVD: cardiovascular disease; MyHeARTs: Malaysian Health and Adolescents Longitudinal Research Team study.

**Figure 3 nutrients-11-00973-f003:**
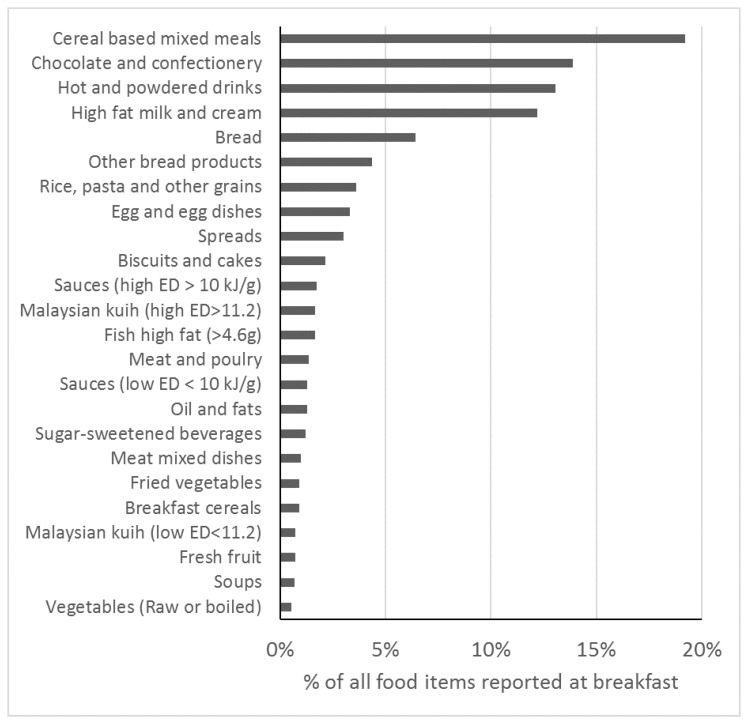
Food groups eaten for breakfast. Bars represent percentage of each food group from all food items reported in breakfast occasions (*n* = 15,507 food items reported at breakfast). Only food groups contributing 1% or more to all food items are displayed. Energy density (ED).

**Figure 4 nutrients-11-00973-f004:**
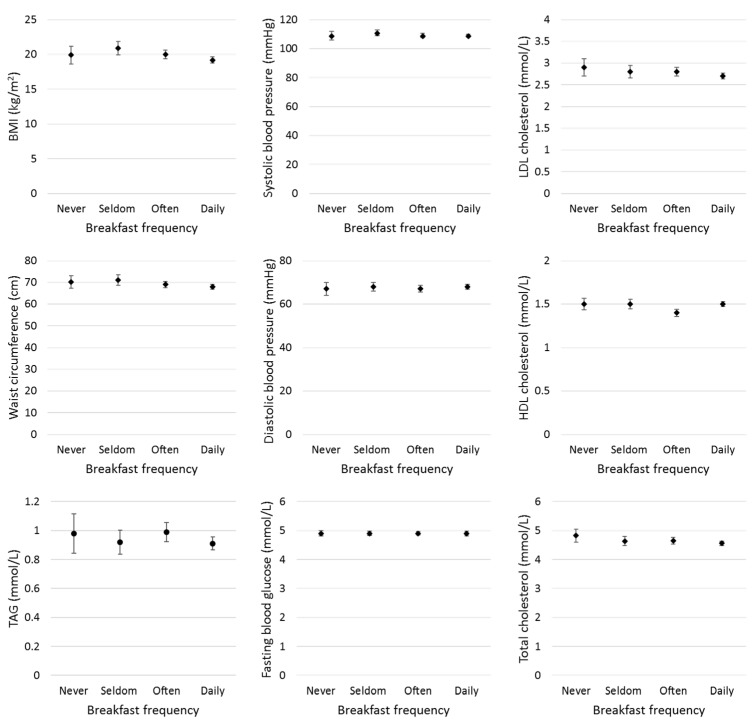
Baseline associations between breakfast frequency and cardiovascular disease (CVD) risk factors in MyHeARTs. Never = 0 days a week, *n* = 76; Seldom = 1–3 days a week, *n* = 115; Often = 4–6 days a week, *n* = 196; Daily = 7 days a week, *n* = 408. Abbreviations—BMI: body mass index; HDL: high-density lipoprotein; LDL: low-density lipoprotein; TAG: triacylglycerol.

**Table 1 nutrients-11-00973-t001:** Characteristics of MyHeARTs participants by breakfast frequency groups.

		Breakfast Frequency				
	Total	0 days/week	1–3 days/week	4–6 days/week	Daily	*p*-Value
All children, n (%)	795 (100)	76 (10)	115 (15)	196 (25)	408 (51)	-
Age (years), mean (SD)	12.9 (0.3)	12.9 (0.3)	12.9 (0.4)	12.8 (0.3)	12.9 (0.3)	-
Sex						
Boys, *n* (%)	298 (37)	32 (10)	36 (12)	58 (19)	172 (58)	0.01 ^a^
Girls, *n* (%)	497 (63)	44 (9)	79 (16)	138 (28)	236 (47)	
Urbanicity						
Urban, *n* (%)	401 (50)	17 (4)	53 (13)	114 (28)	217 (54)	<0.001 ^a^
Rural, *n* (%)	394 (50)	59 (15)	62 (16)	82 (21)	191 (48)	
Ethnicity						
Malay, *n* (%)	650 (82)	68 (10)	100 (15)	162 (25)	320 (49)	0.03 ^a^
Chinese, *n* (%)	44 (6)	0 (0)	8 (18)	12 (27)	24 (55)	
Indian, *n* (%)	69 (14)	5 (7)	3 (4)	12 (17)	49 (71)	
Others, *n* (%)	32 (6)	3 (9)	4 (13)	10 (31)	15 (47)	
Smoking status						
Yes, *n* (%)	73 (9)	11 (15)	12 (16)	13 (18)	37 (51)	0.23 ^a^
No, *n* (%)	722 (91)	65 (9)	103 (14)	183 (25)	371 (51)	
Alcohol intake						
Yes, *n* (%)	23 (3)	3 (13)	5 (22)	4 (17)	11 (48)	0.55 ^a^
No, *n* (%)	772 (97)	73 (9)	110 (14)	192 (25)	397 (51)	
Physical activity (in last 7 days)						
Never n (%)	237 (30)	27 (11)	35 (15)	55 (23)	120 (51)	0.35 ^a^
1–2 times last week *n* (%)	361 (45)	32 (8)	58 (16)	89 (25)	182 (50)	
3–4 times last week *n* (%)	113 (14)	8 (7)	13 (12)	31 (27)	61 (54)	
5–6 times last week *n* (%)	34 (4)	2 (6)	8 (24)	8 (24)	16 (47)	
7+ times last week *n* (%)	50 (6)	7 (14)	1 (2)	13 (26)	29 (58)	
Total daily intake						
Energy (kcal/day)	1673 (332)	1744 (433)	1573 (276)	1594 (267)	1726 (339)	<0.001 ^b^
Protein (% of total energy)	15 (2)	15 (2)	15 (2)	15 (2)	15 (2)	0.63 ^b^
Fat (% of total energy)	29.930 (5)	30 (5)	30 (4)	30 (5)	30 (4)	0.87 ^b^
Carbohydrate (% of total energy)	55 (5)	55 (5)	56 (5)	55 (5)	55 (6)	0.62 ^b^
Cholesterol (mg/1000 kcal)	133 (52)	137 (58)	139 (50)	131(47)	132 (54)	0.53 ^b^
SFA (% of total energy)	6 (2)	6 (2)	5 (2)	6 (3)	6 (2)	0.14 ^b^
Sodium (mg/1000 kcal)	1387 (345)	1328 (380)	1363 (352)	1400 (329)	1399 (344)	0.32 ^b^
Calcium (mg/1000 kcal)	226 (91)	229 (116)	195 (827)	222 (81)	236 (92)	<0.001 ^b^
Iron (mg/1000 kcal)	9 (3)	9 (3)	8 (6)	8 (2)	9 (2)	0.42 ^b^
Crude fiber (g/1000 kcal)	2 (1)	2 (1)	2 (1)	2 (11)	2 (1)	0.51 ^b^
Sugar (% of total energy)	8 (4)	8 (4)	7 (3)	9 (4)	8 (4)	0.04 ^b^
Dietary pattern score (SD units)	0.01 (1.10)	−0.06 (1.18)	−0.09 (1.02)	−0.12 (1.081)	0.12 (1.10)	0.05 ^b^

^a^ Pearson’s chi-squared; ^b^ ANOVA. Abbreviations—ANOVA: analysis of variance; SD: standard deviation; SFA: saturated fatty acid.

**Table 2 nutrients-11-00973-t002:** Description of cardiovascular disease (CVD) risk factors by breakfast frequency (*n* = 795).

	Breakfast Frequency				*p*-Value (ANOVA)	*p* _trend_
	0 days/week (n = 76)	1–3 days/week (*n* = 115)	4–6 days/week (*n* = 196)	Daily (*n* = 408)		
	Mean (SD)	Mean (SD)	Mean (SD)	Mean (SD)		
BMI (kg/m^2^)	19.9 (5.7)	20.9 (5.3)	20.0 (4.4)	19.2 (4.7)	0.003	0.06
WC (cm)	70.1 (12.9)	71.0 (13.4)	69.0 (10.6)	67.9 (11.6)	0.06	0.91
FBG (mmol/L)	4.9 (0.4)	4.9 (0.4)	4.9 (0.4)	4.9 (0.8)	0.79	0.61
TC (mmol/L)	4.8 (1.0)	4.6 (0.9)	4.7 (0.8)	4.6 (0.8)	0.01	0.32
HDL(mmol/L)	1.5 (0.3)	1.5 (0.3)	1.4 (0.3)	1.5 (0.3)	0.56	0.38
LDL (mmol/L)	2.9 (0.9)	2.8 (0.8)	2.8 (0.7)	2.7 (0.7)	0.01	0.41
SBP (mmHg)	109 (13)	111 (11)	109 (11)	109 (11)	0.32	0.31
DBP (mmHg)	67 (13)	68 (10)	67 (10)	68 (10)	0.45	0.34
TAG (mmol/L)	0.98 (0.60)	0.92 (0.45)	0.99 (0.47)	0.91 (0.46)	0.24	0.23

Abbreviations—ANOVA: analysis of variance; BMI: body mass index; DBP: diastolic blood pressure; FBG: fasting plasma glucose; HDL: high-density lipoprotein; LDL: low-density lipoprotein; TC: total cholesterol; SBP: systolic blood pressure; SD: standard deviation; TAG: triacylglycerol; WC: waist circumference.

**Table 3 nutrients-11-00973-t003:** Multiple linear regression analysis of association of breakfast frequency and risk factors of cardiovascular disease independent of potential confounding among participants (*n* = 795).

	β (95% CI)	*p*-Value
Body mass index (kg/m^2^)		
Model 1 ^a^	−0.21 (−0.35, −0.07)	0.004
Model 2 ^b^	−0.20 (−0.34, −0.06)	0.01
Model 4 ^d^	−0.20 (−0.34, −0.05)	0.01
Model 5 ^f^	−0.18 (−0.33, −0.04)	0.01
Waist circumference (cm)		
Model 1 ^a^	−0.44 (−0.78, −0.10)	0.01
Model 2 ^b^	−0.50 (−0.84, −0.16)	0.004
Model 3^c^	−0.06 (−0.21, 0.09)	0.40
Model 4 ^e^	−0.07 (−0.22, 0.08)	0.36
Model 5 ^g^	−0.06 (−0.21, 0.09)	0.43
Fasting glucose concentration (mmol/L)		
Model 1 ^a^	0.00 (−0.02, 0.02)	0.80
Model 2 ^b^	0.00 (−0.02, 0.02)	0.88
Model 3 ^c^	0.00 (−0.02. 0.02)	0.79
Model 4 ^e^	0.00 (−0.02, 0.02)	086
Model 5 ^g^	0.00 (−0.02, 0.02)	0.82
Total cholesterol concentration (mmol/L)		
Model 1 ^a^	−0.04 (−0.06, −0.01)	0.004
Model 2 ^b^	−0.04 (−0.06, −0.01)	0.003
Model 3 ^c^	−0.04 (−0.06, −0.01)	0.01
Model 4 ^e^	−0.03 (−0.06, −0.01)	0.01
Model 5 ^g^	−0.03 (−0.06, −0.01)	0.01
High-density lipoprotein cholesterol concentration (mmol/L)		
Model 1 ^a^	0.00 (−0.01, 0.01)	0.68
Model 2 ^b^	0.00 (−0.01, 0.01)	0.66
Model 3 ^c^	0.00 (−0.01, 0.01)	0.57
Model 4 ^e^	0.00 (−0.01, 0.01)	0.80
Model 5 ^g^	0.00 (−0.01, 0.01)	0.82
Low-density lipoprotein cholesterol concentration (mmol/L)		
Model 1 ^a^	−0.03 (−0.06, −0.01)	0.002
Model 2 ^b^	−0.04 (−0.06, −0.01)	0.001
Model 3 ^c^	−0.03 (−0.05, −0.01)	0.03
Model 4 ^e^	−0.03 (−0.05, −0.01)	0.01
Model 5 ^g^	−0.03 (−0.05, −0.01)	0.01
Systolic blood pressure (mmHg)		
Model 1 ^a^	−0.18 (−0.51, 0.15)	0.28
Model 2 ^b^	−0.14 (−0.46, 0.18)	0.389
Model 3 ^c^	0.04 (−0.25, 0.34)	0.78
Model 4 ^e^	0.07 (−0.23, 0.37)	0.65
Model 5 ^g^	0.07 (−0.23, 0.37)	0.65
Diastolic blood pressure (mmHg)		
Model 1 ^a^	0.11 (−0.19, 0.41)	0.48
Model 2 ^b^	0.12 (−0.18, 0.42)	0.44
Model 3 ^c^	0.26 (−0.03, 0.55)	0.08
Model 4 ^e^	0.27 (−0.03, 0.56)	0.08
Model 5 ^g^	0.27 (−0.02, 0.57)	0.07
Triacylglycerol concentration (mmol/L)		
Model 1 ^a^	−0.01 (−0.02, 0.01)	0.26
Model 2 ^b^	−0.01 (−0.02, 0.01)	0.34
Model 3 ^c^	0.00 (−0.01, 0.01)	0.92
Model 4 ^e^	0.00 (−0.02, 0.01)	0.82
Model 5 ^g^	0.00 (−0.01, 0.01)	0.90

Abbreviations—CI: confidence interval. ^a^ Model 1 includes only breakfast frequency. ^b^ Model 2 includes breakfast frequency, physical activity, smoking status, alcohol intake, ethnicity, and sex. ^c^ Model 3 includes variables in model 2 plus BMI. ^d^ Model 4 includes variables from model 2 plus total daily energy (kcal), total daily carbohydrate (%), total daily protein (%), total daily fat (%), total daily cholesterol (mg/1000 kcal), total daily saturated fatty acid (%), total daily sodium (mg/1000 kcal), total daily calcium (mg/1000 kcal), total daily iron (mg/1000 kcal), total daily fiber (g/1000 kcal), and total daily sugar (%). ^e^ Model 4 includes variables from model 3 plus total daily energy (kcal), total daily carbohydrate (%), total daily protein (%), total daily fat (%), total daily cholesterol (mg/1000 kcal), total daily saturated fatty acid (%), total daily sodium (mg/1000 kcal), total daily calcium (mg/1000 kcal), total daily iron (mg/1000 kcal), total daily fiber (g/1000 kcal), and total daily sugar (%). ^f^ Model 5 includes variables from model 4 ^d^ plus dietary pattern score. ^g^ Model 5 includes variables from model 4 ^e^ plus dietary pattern score.

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
