# Peer review of "The Association of Breakfast Frequency and Cardiovascular Disease (CVD) Risk Factors among Adolescents in Malaysia"

_nutrients, 2019, doi:10.3390/nu11050973_

Round 1
Reviewer 1 Report
The manuscript examines breakfast-CDV association in Malaysian adolescents.
The introduction requires extensive revision. As it stands does not clearly explain what is the novelty of the research, why there has been a need for this research, etc.
There are also sections that have reported contradictory information.
Please find attached PDF with some of my suggestions/comments.

Author Response
Reviewer 1:
[Reviewer comment] The introduction requires extensive revision. As it stands does not clearly explain what is the novelty of the research, why there has been a need for this research, etc.
[Reviewer comment] There are also sections that have reported contradictory information.
[Reviewer comment] Please find attached PDF with some of my suggestions/comments.
[Author response] Thank you for your comments. We have revised the manuscript abstract and introduction to more clearly outline the novelty of the research and address any contradictory information. Please see below the specific edits we have made in response to the comments made inn your annotated PDF.
[Reviewer comment] Line 16-17: “Associations of breakfast with CVD risk in Malaysia are unknown but the type of food consumed at breakfast differs from Western diets so the mechanisms underlying breakfast-CVD risk associations can be explored.” this sentence requires revision. "but" doesnt make sense in the context of this sentence.
[Author response] We have revised the text and hope both the novelty of the study and the meaning of the sentence are now clearer. Line 15-19 now reads “Breakfast frequency is associated with cardiovascular disease (CVD) risk in Western populations, possibly via the types of food eaten or the timing of food consumption, but associations in Malaysian adolescents are unknown. While the timing of breakfast is similar the type of food consumed at breakfast in Malaysia differs from Western diets, which allows novel insight into the mechanisms underlying breakfast and CVD risk.”
[Reviewer comment] Line 28-29: “After adjustment, each day per week that breakfast was eaten was associated with a lower BMI” this sentence is not clear and must be revised.
[Author response] We have changed this sentence to read and hope it is now clearer “After adjustment, each extra day of breakfast consumption per week was associated with a lower BMI” (line 30-31)
[Reviewer comment] Line 53: “breakfast frequency” in previous sentences, kinds of foods for breakfast is the main point of discussion, but not in this sentence. This sentence must be revised.
[Author response] Thanks for highlighting the lack of clarity in our communication of ideas. The start of the paragraph highlights 50 cross-sectional studies of breakfast skipping, in these studies the measurement of skipping breakfast is often simply measured as a response to the question “How often do you eat breakfast?”, hence this is primarily an indicator of breakfast frequency and not the type of food eaten for breakfast. Figure 1 highlights that food choices are just one possible mechanism via which eating breakfast might be linked to cardiovascular disease risk. The other two mechanisms do not relate to the type of food eaten but the impact that eating anything in the morning has on concurrent metabolism and subsequent food intake over the rest of the day. We have expanded the explanation and hope it is now clearer “Several causal mechanisms have been hypothesized to explain why eating breakfast may protect against CVD risk (Figure 1). In pathway 1, eating breakfast is hypothesized to reduce subsequent snacking resulting in lower overall daily energy intake, thus maintaining energy balance and a lower body weight. Although in observational studies, breakfast is typically associated with higher total daily energy intakes [7]. In pathway 2, breakfast is associated with better food choices. For example, the kinds of foods typically eaten at breakfast in the US and Europe tend to have cardio-protective properties e.g. wholegrain cereals are high in fiber and micronutrients [8]. However, as foods typically eaten for breakfast vary widely across cultures, consistent associations of breakfast frequency across diverse countries could point to the timing of eating as more important than the type of food eaten. In pathway 3, it’s hypothesized that eating in the morning is specifically better suited to circadian rhythms in metabolism such that food ingested earlier in the day is metabolized more efficiently. The hypothesised circadian pathway is supported by acute randomised randomized crossover experiments in adults, whereby the timing of food intake (but not the type of food consumed) is manipulated such that delaying breakfast is related to poorer appetite control, lower resting energy expenditure, impaired fasting lipid profiles and reduced postprandial insulin sensitivity [9-12].” (line 53-67)
[Reviewer comment] Line 54-55: “as more important. Acute randomised crossover experiments in adults, whereby the timing of food intake is manipulated” what does this sentence mean? It is not clear and must be revised
[Author response] We have revised the sentence and hope that the meaning is now clearer “In pathway 2, breakfast is associated with better food choices. For example, the kinds of foods typically eaten at breakfast in the US and Europe tend to have cardio-protective properties e.g. wholegrain cereals are high in fiber and micronutrients [8]. However, as foods typically eaten for breakfast vary widely across cultures, consistent associations of breakfast frequency across diverse countries could point to the timing of eating as more important than the type of food eaten.” (line 56-61)
[Reviewer comment] Line 57-58: “delaying the timing of the first meal or simple advice to “eat breakfast” these 2 examples are contradictory. What do authors try to explain here? This must be revised.
[Author response] Thanks for highlighting the issue, we have reworded for clarity “However more recent longer-term free-living trials delaying that manipulated the timing of the first meal or randomized participants to receive simple advice to “eat breakfast” or not, have cast doubt on whether observational associations are causal but instead confounded i.e. eating breakfast may simply be an indicator of a generally healthy lifestyle [13-15]” (line 68-71)
[Reviewer comment] Line 68-69: “Considering the latter two phenomena can be influenced by breakfast consumption,” requires reference
[Author response] We have added the reference [19] in line 82 doi: https://doi.org/10.2337/dc16-2753 “Influences of Breakfast on Clock Gene Expression and Postprandial Glycemia in Healthy Individuals and Individuals With Diabetes: A Randomized Clinical Trial” to support our statement.
[Reviewer comment] Line 70: “adolescents” The population of interest hasn't been clearly defined through the manuscript. Adolescents and children are different, with different characteristics. This needs to be defined clearly.
[Author response] We have added to line 78, so it reads “Most research on breakfast and health has been conducted in adults, with some work in children, but far less research has been dedicated to adolescents. However, adolescence represents a unique phase of life involving rapid growth, hormonal fluctuations, insulin resistance[16, 17], and circadian dysregulation[18].” The following sentence goes on to explain why adolescents are unique.
[Reviewer comment] Line 83-84: “Given that higher adiposity is associated with poorer CVD risk profiles, one might expect to see similar associations with breakfast frequency.” how this conclusion can be made? what is the evidence/reference?
[Author response] We have added the reference [27] line doi: 10.21037/atm.2017.03.107 “Morbidity and mortality associated with obesity” to demonstrate the relationship between obesity and CVD risk factors. We have also edited the sentence for complete clarity to read “Given that higher adiposity is associated with poorer CVD risk profiles [27], and breakfast frequency is associated with adiposity, one might expect to see similar associations between breakfast frequency and cardiovascular risk.”
[Reviewer comment] Line 83-84: “Finally, a proposed method for improved understanding of causality in observational research is to explore the consistency of associations across high, middle- or low-income countries where the confounding structure may differ[34].” how causality can be examined through observational studies??
[Author response] Thank you for this comment. Observational studies have limitations for assessing causality, for example confounding and reverse causation, but combining the findings of different study designs can improve causal inference via triangulation. According to Bradford Hill multiple criteria should be satisfied to conclude that a given factor is causally related to another and a single study (especially an observational one) will be limited in its ability to conclude cause and effect. In the citation provided [37] there is further explanation regarding how studying the same phenomena in different cultural contexts where the confounding structure differs can improve on a single observational study. The more consistently associations are observed, despite different sources of confounding the more likely it is that the factor may be causal. This does not preclude the need for further confirmatory evidence from experiments but does add informatively to the body of evidence. In Malaysia breakfast consumption may be associated with socio-economic status, physical activity or ethnicity in ways that differ from confounding structures seen previous literature primarily based in Western countries. Thus if an association between breakfast and CVD risk holds up across multiple contexts, conclusions can be strengthened as it is less likely to be the result of confounding.
[Reviewer comment] Line 112: “Thus” what is the novelity of this study?
[Author response] We have added further explanation to highlight the novelty of the study: “No study to date has investigated the association of breakfast and CVD risk in Malaysian adolescents. In addition, few previous studies are able to explore what is eaten for breakfast in addition to breakfast frequency.” (line 129-131)
[Reviewer comment] Line 113: “investigate” ?
[Author response] Thanks for highlighting this typo we have edited the sentence to now read “The purpose of this our study was to provide a detailed description of what adolescents ate for breakfast and investigate the association of breakfast frequency with CVD risk factors among adolescents using data from the Malaysian Health and Adolescent Longitudinal Research study (MyHeARTs), including 7-day diet histories enabling adjustment for potential confounders including concurrent diet quality.” (line 131-135)
Reviewer 2 Report
Thank you for the interesting article. Minor changes in the English language is required. Especially some sentences in line 295-300 seem to be complicated. They can be reconstructed.
Some spellings can be checked and rewritten in American English (e.g., characterized). In the discussion, subheadings can be used (the result of the study, review of the available data, unique highlights of this study, limitation if any, conclusion).
Author Response
We would like to thank the
reviewers for their time reviewing and giving feedback on our manuscript. We
have addressed the comments as appropriate, with our responses to comments
below (in blue), and a revised manuscript resubmitted with track changes.
Reviewer 2:
[Reviewer comment] Thank you for the interesting article. Minor changes in the English language is required. Especially some sentences in line 295-300 seem to be complicated. They can be reconstructed.
[Author response] We have edited lines 295-300 as suggested “For example, some studies defined breakfast based on the time of the day food was consumed, e.g. between 0600 0900h [46], while others based their definitions on the frequency of breakfast (self-identified by the participant) during the week [29, 32, 34] or only during weekdays [44]. In addition, adolescents with the same breakfast frequency of may be in different groups across studies, e.g. breakfast skippers in the current study are those who never consumed breakfast, while in other studies those who never consumed breakfast are combined with irregular consumers [29].” (line 314-320)
[Reviewer comment]Some spellings can be checked and rewritten in American English (e.g., characterized).
[Author response] We have edited throughout to conform to American English.
[Reviewer comment] In the discussion, subheadings can be used (the result of the study, review of the available data, unique highlights of this study, limitation if any, conclusion).
[Author response] Thank you for this suggestion. We have included subheadings in the discussion.
Round 2
Reviewer 1 Report
The MS has improved. However, revision is required, especially in the conclusion, before consideration for publication:
1. There are a number of typo through the MS. Please revise accordingly.
2. Also, in line 477, do authors mean "lower CVD risk factors"? Please revise accordingly.
3. The conclusion needs to be ellaborated further. What would be the benefit of longitudinal study for this population and what would be potential suggestions? How findings from this research can be used to design a longitudinal study?
"Future research should aim to examine whether daily breakfast intake is associated with lower CVD risk both longitudinally and causally in adolescents."
Author Response
We would like to thank the reviewer for their additional time giving feedback on our manuscript. We have addressed the comments as appropriate, with our responses to comments below (in red), and a revised manuscript resubmitted with track changes.
[Reviewer comment] The MS has improved. However, revision is required, especially in the conclusion, before consideration for publication:
[Author response] Thank you, we are glad that our revisions addressed your previous comments and appreciate you taking the time to provide further comments for revision.
[Reviewer comment] 1.There are a number of typo through the MS. Please revise accordingly.
[Author response] Thank you for highlighting this, we have reviewed the entire manuscript and have identified and corrected all typos, please see below:
Line 34: ‘,’ replaced with ‘.’
Line 87: removed ‘
Line 92: olds à old
Line 103: added ‘a’
Line 111: 16 year olds à 16 years-olds
Line 128: added ‘of’
Line 134: histories à history
Line 143: obtain à obtained
Line 169: a à an
Line 208: added space between version 15, StataCorp
Line 225: foods à food
Line 234: Characteristic à Characteristics
Line 248: maybe à may be, breakfasts à breakfast
Line 284: added ‘was’
Line 287: model à Model, table à Table
Line 310: year olds à year-olds
Line 319: removed ‘of’
Line 379: adolescents à adolescence
[Reviewer comment] 2. Also, in line 477, do authors mean "lower CVD risk factors"? Please revise accordingly.
[Author response] Thanks for highlighting the lack of clarity. We have added to line 474-476 so that it now reads “Thus, both longitudinal and randomised randomized controlled studies are required in order to establish the causality of the relationship between breakfast consumption and lower CVD risk factors in this population.”
[Reviewer comment] 3. The conclusion needs to be ellaborated further. What would be the benefit of longitudinal study for this population and what would be potential suggestions? How findings from this research can be used to design a longitudinal study?
"Future research should aim to examine whether daily breakfast intake is associated with lower CVD risk both longitudinally and causally in adolescents."
[Author response] We have expanded the conclusion as follows “In conclusion, the present study suggests that adolescents in Malaysia who eat breakfast more frequently have a lower BMI, and lower plasma total and LDL cholesterol concentrations independent of a range of confounders. Our study results specifically highlight that the benefits of breakfast (if causal) are not limited to the types of foods typically eaten in Western populations such as breakfast cereals, milk, or bread but may be independent of the type of food consumed and instead related to the timing of food intake. Furthermore, our data suggested that irregular habits may be more detrimental to CVD risk that uniform breakfast eating vs. skipping. Previous longitudinal studies or long-term randomized trials have not explored the potential impact of consistent vs. irregular breakfast habits or been able to differentiate changes in the timing of eating from the type of food eaten. Future research should aim to examine whether the content, timing or regularity of daily breakfast intake is associated with lower CVD risk both longitudinally and causally within trials in adolescents.”